# Radiopathological Correlation in Locally Advanced Rectal Cancer After Neoadjuvant Treatment

**DOI:** 10.3390/cancers17243937

**Published:** 2025-12-09

**Authors:** Mario Martín-Sánchez, Pedro Villarejo Campos, Víctor Domínguez-Prieto, Eva Ruiz-Hispán, Begoña López-Botet Zulueta, Carlos Pastor, Miguel León-Arellano, Héctor Guadalajara, Damián García-Olmo, Siyuan Qian-Zhang

**Affiliations:** 1Surgery Department, Hospital Universitario Fundación Jiménez Díaz, Avenida de los Reyes Católicos 2, 28040 Madrid, Spainsiyuanqz@gmail.com (S.Q.-Z.); 2Surgery Department, Universidad Autónoma de Madrid, 28049 Madrid, Spain; 3Oncology Department, Hospital Universitario Fundación Jiménez Díaz, Avenida de los Reyes Católicos 2, 28040 Madrid, Spain; 4Radiology Department, Hospital Universitario Fundación Jiménez Díaz, Avenida de los Reyes Católicos 2, 28040 Madrid, Spain; 5Surgery Department, Clínica Universitaria de Navarra, C. del Marquesado de Sta, Marta, 1, San Blas-Canillejas, 28027 Madrid, Spain

**Keywords:** locally advanced rectal cancer, magnetic resonance imaging, tumor regression grade, neoadjuvant chemoradiotherapy, organ preservation

## Abstract

Accurate assessment of tumor response after neoadjuvant treatment is essential for the management of patients with locally advanced rectal cancer. This study evaluated the correlation between MRI-based tumor regression grade and the histopathological tumor regression grade observed in surgical specimens. The results revealed only a fair correlation between MRI findings and pathological outcomes, indicating that MRI alone may be insufficient to fully assess treatment response. However, MRI demonstrated high specificity, suggesting it can reliably confirm when patients have achieved a favorable response. These findings highlight the importance of integrating MRI with complementary diagnostic modalities to improve the precision of treatment response evaluation and to guide personalized therapeutic decisions.

## 1. Introduction

Colorectal cancer (CRC) is the third most prevalent cancer worldwide, with rectal cancer accounting for approximately 729,833 cases globally in 2022. Notably, the incidence of rectal cancer among individuals under 50 years of age has been increasing, representing a concerning epidemiological trend [1].

The primary diagnostic modalities for staging rectal cancer are magnetic resonance imaging (MRI), computed tomography (CT) scan and endorectal ultrasound [2]. Although CT scan is valuable for detecting distant metastases, it has limited spatial resolution and reduced accuracy in evaluating tumors located in the mid and lower rectum. Endorectal ultrasound is precise for detecting T1–T2 tumors but has significant limitations in T3–T4 due to a limited field of view and a strong dependence on operator expertise [3]. MRI is considered the cornerstone of rectal cancer staging, offering high-resolution images and precise anatomical detail that are critical for determining the optimal treatment strategy [3,4,5,6].

Currently, the standard treatment approach for locally advanced rectal cancer (LARC), defined as stage T3–T4 or any T stage with regional lymph nodes involvement, consists of neoadjuvant chemoradiotherapy (CRT) followed by surgery and adjuvant chemotherapy. This multimodal strategy has demonstrated good oncological results in terms of local recurrence (LR) but not in overall survival (OS) due to the elevated risk of systemic relapse [2,7].

Recently, total neoadjuvant therapy (TNT) has emerged as a promising alternative, demonstrating improved survival and higher rates of pathological complete response (pCR) [8,9,10,11,12]. Therefore, in recent years, there has been a growing interest in the watch-and-wait strategy as an alternative to immediate surgery in patients with a complete clinical response, offering selected patients with great response to chemoradiotherapy the opportunity to avoid unnecessary procedures and potentially maintain a better quality of life [13,14,15]. Regardless of the chosen treatment approach, accurate patient selection for this strategy and meticulous follow-up are essential.

Assessing tumor response after neoadjuvant therapy remains a significant challenge for radiologists due to post-treatment changes such as fibrosis, mucosal edema and rectal wall thickening, which can mimic or obscure residual disease [16,17,18,19].

To address this, the magnetic resonance tumor regression grade (mrTRG) system was developed to stratify tumor response based on MRI findings, following a conceptual framework similar to the pathological tumor regression grade (pTRG), with the aim of guiding clinical decision-making without the need for a surgical specimen [20,21,22]. A key clinical consideration is understanding the diagnostic reliability and utility of MRI in predicting pathological tumor response in routine practice. While mrTRG systems have been developed to guide clinical decision-making, the performance characteristics and limitations of MRI warrant careful examination, particularly in the current era of organ preservation strategies where imaging interpretation directly influences patient management and treatment selection.

Although a complete clinical response (cCR) is often assumed to reflect a pCR, the correlation between mrTRG and pTRG following neoadjuvant CRT has not been well validated.

This study aims to investigate the correlation between MRI findings and pathological results in patients with LARC who have undergone neoadjuvant CRT and subsequent surgery to establish its role in evaluating tumor response after treatment. Additionally, this research seeks to identify any potential discrepancies between imaging and pathology results, providing valuable insights into the challenges and limitations of current radiological evaluations.

## 2. Materials and Methods

### 2.1. Patients

A retrospective analysis was conducted on 97 patients diagnosed with LARC at our hospital (Hospital Universitario Fundación Jiménez Díaz) from 2014 to 2020. Patients were diagnosed and staged through MRI, endorectal ultrasound and CT-scan in accordance with the National Comprehensive Cancer Network (NCCN) guidelines [23]. Consistent with standard practice in retrospective institutional cohort studies, no a priori sample size calculation was performed; all consecutive eligible patients meeting the inclusion criteria during the study period were included, ensuring a comprehensive representation of this patient population.

To ensure appropriate patient selection and consistent data quality, the following inclusion and exclusion criteria were applied.

#### 2.1.1. Inclusion Criteria

-Patients diagnosed with middle or low rectal cancer, defined as tumors located within 10 cm from the anal verge.-Completion of neoadjuvant CRT according to institutional and guideline-based protocols.-Availability of high-quality MRI studies performed both prior to and following neoadjuvant treatment for accurate radiological assessment.-Patients who subsequently underwent surgical resection with histopathological evaluation of the surgical specimen.

#### 2.1.2. Exclusion Criteria

-Patients who did not complete the prescribed neoadjuvant chemoradiotherapy protocol.-Absence of adequate pre-treatment or post-treatment MRI studies, or MRI scans of insufficient quality for reliable assessment.-Tumors located proximal to 10 cm from the anal verge (i.e., outside the middle or low rectum).-Contraindications to MRI scanning, including but not limited to comorbidities or implanted medical devices incompatible with MRI.-Patients who underwent palliative surgery or those without available surgical specimens for histopathological analysis.

### 2.2. Neoadjuvant Chemoradiotherapy

Based on our hospital protocol and according to NCCN and European Society for Medical Oncology (ESMO) guidelines, all patients underwent modern modulated radiotherapy delivered using either intensity-modulated radiotherapy (IMRT) or volumetric modulated arc therapy (VMAT). Patients were treated with long-course radiotherapy delivering 50.0–50.4 Gy in 25–28 fractions, administered on consecutive weekdays. Image-guided radiotherapy (IGRT) with daily cone-beam CT (CBCT) verification was systematically employed following a planning CT performed under bladder and bowel preparation. All patients received concurrent chemotherapy with capecitabine at 825 mg/m^2^ twice daily during the radiotherapy regimen [19,20].

### 2.3. MRI: Staging and Reevaluation After Neoadjuvant Treatment

Rectal MRI studies were performed before and after treatment using 1.5 and 3.0 T scanners with 2D T2-weighted sequences in the sagittal, axial, high-resolution oblique axial, and coronal planes, and axial diffusion-weighted imaging (DWI), with approximately 150 mL of intrarectal ultrasound gel. No previous rectal preparation, spasmolytic, or IV contrast was used.

MrTRG assessments were performed by a gastrointestinal radiologist with more than five years of experience. No inter-observer or intra-observer reproducibility analysis was performed.

The response to neoadjuvant therapy was evaluated according to mrTRG (grades 1–5) established in MERCURY study (Table 1) [22].

### 2.4. Histopathological Examination

Expert pathologists examined surgical specimens. In addition to the standard assessment, the tumor regression grade was evaluated following the Mandard classification (Table 2) [20].

As it was assumed in previous studies, a good response is established as pTRG1–2 and mrTRG1–2 [24]. In addition, pTRG1 or mrTRG1 is considered a complete response.

### 2.5. Statistical Analysis

The main objective of this study is to evaluate the correlation between mrTRG and pTGR. To achieve this, we calculated the kappa value and the weighted Kappa test.

This statistical tool is interpreted as follows: 1–0.81 excellent agreement; 0.80–0.61 good agreement; 0.60–0.41 moderate agreement; 0.4–0.21 fair agreement; 0.2–0.00 poor agreement (Table 3).

To assess sensitivity and specificity, a dichotomous classification was applied: good response (mrTRG 1–2 and pTRG 1–2) or complete response (mrTRG 1 and pTRG 1), compared with poor response (mrTRG 3–5) or incomplete response (mrTRG 2–5).

Receiver operating characteristic (ROC) curves and area under the curve (AUC) values were calculated to assess the discriminative ability of mrTRG to predict both good pathological response (pTRG 1–2) and complete response (pTRG 1). The AUC is interpreted as follows: 0.90–1.00, excellent; 0.75–0.89, good; 0.60–0.74, modest; 0.50–0.59, poor. An AUC of 0.50 indicates no discriminative ability.

These analyses were performed using SPSS version 25 (IBM Corp., Armonk, NY, USA) and R version 4.2.0.

## 3. Results

### 3.1. Patients and Tumor Characteristics

A total of 97 patients were included in the analysis, 57 males (58.8%) and 40 females (41.2%), with a median age of 67 years (range 37–88 years). The median tumor distance from the anal verge was 7.6cm (range 1–10 cm). Circumferential resection margin (CRM) was affected in 38 patients (39.2%). According to the 7th edition of the American Joint Committee on Cancer (AJCC) TNM classification system, 77 patients (79.4%) were classified as T3, while 20 (20.6%) were classified as T4. Regarding nodal status, 49 (50.5%) patients were identified as N1 and 48 (49.5%) as N2.

Regarding surgical treatment, a total mesorrectal excision was achieved in 74 patients (76.3%). The median number of harvested lymph nodes was 15.43 (range 5–29). The median days of admission was 11 days. All parameters are summarized in Table 4.

### 3.2. Time Intervals Between Neoadjuvant Chemoradiotherapy, Restaging MRI and Surgery

The median interval from completion of neoadjuvant chemoradiotherapy and restaging MRI was 40 days (range: 8–96). The median time between MRI and radical surgery was 29 days (range: 2–124), resulting in a total median interval of 69 days (range: 32–163) from completion of CRT to surgery (Figure 1).

### 3.3. Association of Initial T and N Stage with pTRG

Based on our data, there is a significant association between the initial T stage and pTRG (*p* < 0.05), suggesting that the T3 stage is more likely to exhibit a good response (pTRG1-2) or a complete response (pTRG1). However, no significant association was observed between the initial N stage and pTRG (*p* = 0.591) (Table 5).

### 3.4. Tumor Response According to mrTRG and pTRG

According to MRI findings, the distribution of mrTRG was: 3 patients (3.1%) with mrTRG1, 31 (32.0%) mrTRG2, 47 (48.5%) mrTRG3, 15 (15.5%) mrTRG4, and 1 (1.0%) mrTRG5. Correspondingly, pTRG using the Mandard classification showed 20 patients (20.6%) as pTRG1, 28 (28.9%) pTRG2, 30 (30.9%) pTRG3, 19 (19.6%) pTRG4, with no cases classified as pTRG5.

A significant association was found between mrTRG and pTRG (*p* = 0.001) (Table 6 and Figure 2).

### 3.5. Sensitivity, Specificity and Kappa Agreement Between mrTRG and pTRG

The weighted Kappa between MRI (mrTRG) and pathologist examination (pTRG) was 0.27, indicating fair agreement.

MRI demonstrated a sensitivity of 52.1% and a specificity of 81.6% in predicting a good response (pTRG 1–2). The positive predictive value (PPV) was 73.5%, and the negative predictive value (NPV) was 63.5%. The kappa coefficient was 0.338, indicating fair agreement.

For the identification of complete response (pTRG 1), sensitivity was 10.0%, while specificity was 98.7%. The PPV and NPV were 66.7% and 80.9%, respectively. The kappa coefficient was 0.127, indicating poor agreement.

To further assess ordinal correlation between the two grading systems, we calculated Spearman’s rank correlation coefficient and Kendall’s τ-b. Spearman’s ρ was 0.413 (*p* < 0.001), and Kendall’s τ-b was 0.367 (*p* < 0.001), confirming a modest positive association between mrTRG and pTRG.

A summary of these diagnostic performance indicators is provided in Table 7.

The ROC curve analysis demonstrated an area under the curve (AUC) of 0.669 for predicting a good response, indicating a modest discriminatory ability of MRI in identifying patients with favorable tumor regression. Conversely, the AUC for predicting a complete pathological response was 0.541, reflecting poor accuracy and limited predictive value of MRI for complete response detection in this cohort (Figure 3).

## 4. Discussion

In this study, we evaluated the correlation between mrTRG and pTRG after neoadjuvant chemoradiotherapy in patients with locally advanced rectal cancer. Our results showed a fair agreement between both classifications (κ = 0.27), with MRI demonstrating high specificity but low sensitivity. These findings suggest that MRI is more reliable in identifying patients who achieve a good or a complete response.

TNT with the intention of organ preservation is already accepted and increasingly integrated into treatment protocols for LARC. MRI remains the primary tool for assessing tumor response after neoadjuvant chemoradiotherapy [2,23,25], a critical step since patients demonstrating a complete response may be candidates to avoid surgery and initiate a watch-and-wait protocol [5,26]. Previous studies have shown that mrTRG 1–2 is associated with improved (OS) and progression-free survival (PFS) in univariate analyses [27,28]. Similarly, pTRG 1–2 has been linked to better oncological outcomes, reinforcing its prognostic relevance [29,30].

The observed low sensitivity and high specificity of MRI in this study carry important clinical implications. Low sensitivity indicates that MRI fails to detect a considerable proportion of true good or complete responses, resulting in false negatives and limiting its value as a standalone tool for ruling out residual disease. Conversely, the high specificity suggests that when MRI identifies a good or complete response, this finding is highly reliable, with a low rate of false positives.

Our findings are consistent with those reported in previous studies. Achilli et al. observed higher sensitivity (38% vs. 10%) and lower specificity (84% vs. 98%) when predicting pathological complete response by MRI, with a slightly higher weighted kappa of 0.386 [31]. Similarly, Sclafani et al. reported a weighted kappa of 0.24, comparable to our results; both studies agree that the concordance between mrTRG and pTRG is limited [21]. More recently, Niu et al. conducted a single-center study involving 54 patients. Their study also found moderate agreement with a weighted kappa of 0.391 for detecting good responses [27].

Despite minor variations in sensitivity, specificity, and kappa values, all studies consistently show a modest correlation between mrTRG and pTRG. Differences may arise from variations in sample size, imaging protocols, and evaluator experience, as well as the multicenter versus single-center design. These factors should be considered when comparing results across studies.

The limited correlation between MRI and pathologist examination may be explained by the challenges in interpreting post-radiotherapy changes, particularly radiation-induced fibrosis, which complicates the distinction between residual tumor and treatment effects. Advanced radiologic techniques such as DWI or apparent diffusion coefficient (ADC) may aid in differentiating fibrosis from viable tumor tissue [3,6].

Furthermore, the dynamic nature of tumor response after chemoradiotherapy should be considered. In our cohort, the average interval between the last MRI and surgery was approximately four weeks, during which ongoing tumor regression may have occurred. Hence, it is plausible that the correlation between radiological and pathological assessments would have been greater if MRI had been performed closer to the time of surgery. Several studies have shown that extending the interval between completion of neoadjuvant therapy and surgery is safe and associated with improved pCR rates [32,33,34]. Notably, intervals exceeding nine weeks have been linked to even better pCR outcomes [35].

Despite its clinical value, MRI alone remains insufficient to accurately determine tumor response following neoadjuvant chemoradiotherapy. Therefore, most watch-and-wait protocols incorporate MRI alongside digital rectal examination and endoscopy to improve the reliability of clinical assessment [36]. New imaging approaches, including artificial intelligence and radiomics, have shown promising results for response prediction in rectal cancer, but remain time-consuming and are not yet widely implemented in clinical practice [3,6,37,38,39]. Our dataset, systematically annotated with both radiological and pathological outcomes, may serve as a useful reference for future external validation studies of such technologies and contribute to the development of more accurate and personalized management algorithms.

Another promising tool in the management of LARC is the liquid biopsy. Recent studies suggest that it may become one of the most relevant prognostic factors in colorectal cancer, with encouraging results in sensitivity, specificity, and predictive value. This non-invasive technique has shown great potential for monitoring tumor dynamics, assessing treatment response, and detecting minimal residual disease [40,41,42,43]. In particular, the detection of *SEPT9* gene hypermethylation one month after surgery has demonstrated excellent performance in identifying tumor recurrence, with reported sensitivity and specificity rates close to 100% and 94.7%, respectively [44]. Its objectivity and reproducibility position it as a potentially valuable decision-making tool in the post-treatment setting.

This study presents several limitations that must be acknowledged. First, its retrospective design may introduce selection and information bias. Second, although the sample size is consistent with previous studies in the field, it remains limited, potentially affecting the statistical power of some analyses. Third, the interval between the final MRI and surgery varied slightly among patients, which may have influenced the correlation between radiological and pathological assessments, as tumor regression may continue after neoadjuvant therapy. In our study, MRI assessment was performed by a single experienced gastrointestinal radiologist with over 5 years of expertise in rectal cancer imaging and pathological evaluation by a single expert pathologist. While this approach eliminates inter-observer variability, it precludes formal assessment of intra- and inter-observer reproducibility. This methodological choice limits the external validity and generalizability of our findings to other centers and radiologists with different experience levels.

## 5. Conclusions

LARC has seen significant advances in the past decade, largely due to the development of more effective neoadjuvant treatments. These have positioned organ preservation as a viable alternative in patients achieving a complete clinical response. Within this context, the need for accurate and reliable diagnostic tools is essential to support treatment decisions and ensure adequate follow-up.

In agreement with previous studies, our findings showed limited concordance between mrTRG and pTRG, indicating that MRI alone may be insufficient for accurate tumor response evaluation. Therefore, clinical assessment by digital rectal examination and endoscopy remains essential. Emerging techniques like radiomics and liquid biopsy hold promise for improving response prediction and personalized treatment, but further research is needed to validate their clinical utility. Nonetheless, the comprehensive nature of our dataset provides a robust contemporary benchmark for future advancements in imaging analytics, such as radiomics and AI-driven prediction models.

Despite its limitations, MRI demonstrated high specificity and negative predictive value, supporting its role as a valuable tool in the prognostic assessment of rectal cancer. Future research should focus on combining MRI with other emerging diagnostic modalities to enhance response evaluation and optimize personalized treatment strategies for patients with locally advanced rectal cancer.

## Figures and Tables

**Figure 1 cancers-17-03937-f001:**
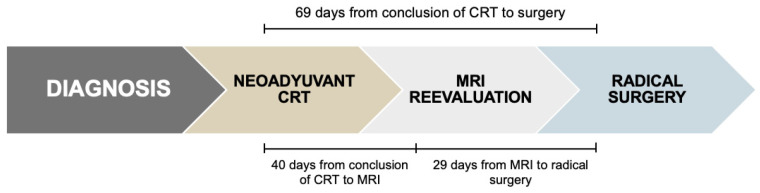
Treatment protocol for LARC.

**Figure 2 cancers-17-03937-f002:**
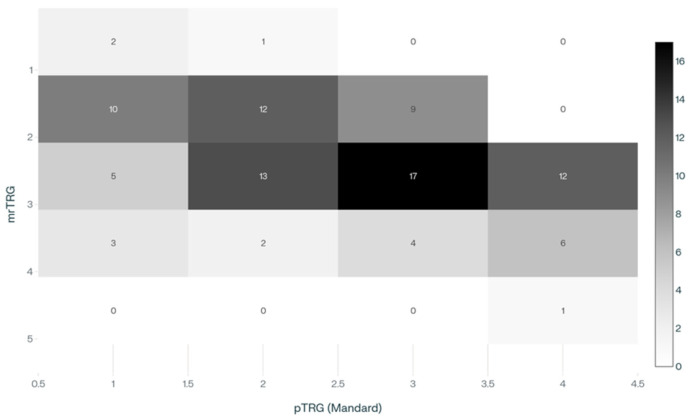
Heatmap showing the distribution of mrTRG ratings across pTRG categories.

**Figure 3 cancers-17-03937-f003:**
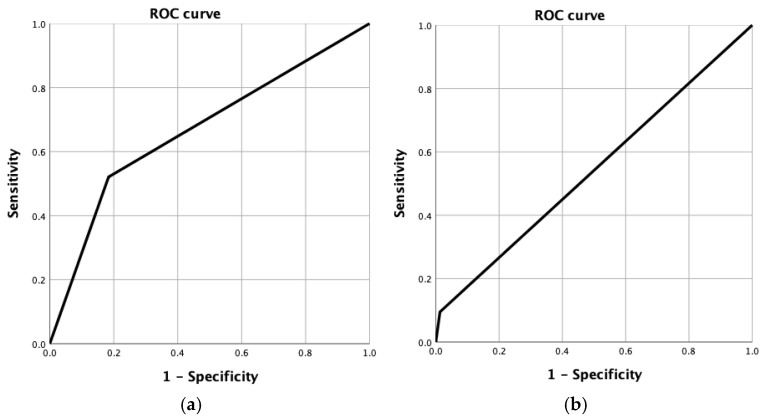
ROC curve for good response (**a**) and complete response (**b**).

**Table 1 cancers-17-03937-t001:** mrTRG classification based on MERCURY study.

**mrTRG1**	No evidence of tumor signal intensity or fibrosis only
**mrTRG2**	Dense fibrosis, minimal residual tumor
**mrTRG3**	Mixed fibrosis/mucin and intermediate signal representing residual tumor, but fibrosis still predominates.
**mrTRG4**	Small areas of fibrosis or mucin but mostly tumor
**mrTRG5**	Same appearance as original tumor or tumor growth

**Table 2 cancers-17-03937-t002:** pTRG classification based on Mandard classification.

**pTRG1**	Absence of residual cancer
**pTRG2**	Presence of residual cancer cells scattered throughout the fibrosis
**pTRG3**	Increase in the number of residual cancer cells but fibrosis still being predominant
**pTRG4**	Residual cancer outgrowing fibrosis
**pTRG5**	Absence of regressive changes

**Table 3 cancers-17-03937-t003:** Interpretation of Kappa Coefficient Values.

**1–0.81**	excellent agreement
**0.80–0.61**	good agreement
**0.60–0.41**	moderate agreement
**0.4–0.21**	fair agreement
**0.2–0.00**	poor agreement

**Table 4 cancers-17-03937-t004:** Patient and tumor characteristics.

Variables	Total (N = 97)
**Age**	67 years (37–88)
**Sex**	
Male	57 (58.8%)
Female	40 (41.2%)
**Distance from anal verge**	7.6cm (1–10)
**CRM affected**	38 (39.2%)
**Total mesorrectal excision**	74 (76,3%)
**Lymph nodes harvested**	15.5 (5–29)
**T stage**	
T3	77 (79.4%)
T4	20 (20.6%)
**N stage**	
N1	49 (50.5%)
N2	48 (49.5%)
**Days of admisions**	11 days

**Table 5 cancers-17-03937-t005:** Cross-tabular analysis between initial T stage and N stage with pTRG (Mandard Classification).

	pTRG1	pTRG2	pTRG3	pTRG4	pTRG5	*p*
**T3**	18	26	21	12	0	0.005
**T4**	2	2	9	7	0	
	**pTRG1**	**pTRG2**	**pTRG3**	**pTRG4**	**pTRG5**	** *p* **
**N1**	10	14	15	10	0	0.591
**N2**	11	14	14	9	0	

**Table 6 cancers-17-03937-t006:** Correlation between mrTRG and pTRG.

	pTRG1	pTRG2	pTRG3	pTRG4	pTRG5	Total	*p*
**mrTRG1**	2 (6.7%)	1 (3.4%)	0 (0%)	0 (0%)	0 (0%)	3 (3.1%)	0.001
**mrTRG2**	10 (33.3%)	12 (40%)	9 (30%)	0 (0%)	0 (0%)	31 (32%)	
**mrTRG3**	5 (16.7%)	13 (43.3%)	17 (56.7%)	12 (63.2%)	0 (0%)	47 (48.5%)	
**mrTRG4**	3 (10%)	2 (6.7%)	4 (13.3%)	6 (31.6%)	0 (0%)	15 (15.5%)	
**mrTRG5**	0 (0%)	0 (0%)	0 (0%)	1 (5.3%)	0 (0%)	1 (1%)	
**Total**	20 (20.6%)	28 (28.9%)	30 (30.9%)	19 (19.6%)	0 (0%)	97 (100%)	

Note: Percentages represent the proportion of each mrTRG category within each pTRG column, showing the distribution of MRI ratings for each pathological response grade.

**Table 7 cancers-17-03937-t007:** Cross-tabular analysis of good and complete response.

	pTRG1-2	pTRG3-5		k
**mrTRG1-2**	25	9	34	0.338
**mrTRG3-5**	23	40	63	
	48	49	97	
	**pTRG1**	**pTRG2-5**		**k**
**mrTRG1**	2	1	3	0.127
**mrTRG2-5**	18	76	94	
	20	77	97	

## Data Availability

The data presented in this study is available on request from the corresponding author. The data is not publicly available due to ethical restrictions.

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
