# Peer review of "Radiopathological Correlation in Locally Advanced Rectal Cancer After Neoadjuvant Treatment"

_cancers, 2025, doi:10.3390/cancers17243937_

Round 1

Reviewer 1 Report

Comments and Suggestions for Authors

Thank you for the opportunity to review this informative study on MRI findings following neoadjuvant treatment for rectal cancer. However, I have several major concerns regarding the study design.

1) Treatment Details: The manuscript lacks specific details regarding the treatment protocols, specifically the radiotherapy parameters (dose/fractionation) and chemotherapy regimens used. As these factors significantly influence tumor response rates, a detailed description is essential for interpreting the results.

2) Observer Variability: The reproducibility and objectivity of the radiologists’ MRI evaluations were not verified. In subjective imaging studies such as this, it is crucial to assess and report intra-observer and inter-observer variability to ensure data reliability.

3) Novelty and Clinical Significance: The study reports a correlation between T2-weighted MRI and pathological findings. However, this is an expected outcome given that tumor shrinkage is a well-documented effect of chemoradiotherapy. The authors have not clearly articulated the novelty of their specific findings. 

4) Statistical Power: The manuscript does not currently include a sample size calculation. Please provide a power analysis or justification for the sample size to demonstrate statistical validity.

Author Response

We thank the reviewer for the thorough and constructive evaluation of our manuscript. The detailed assessment of the study design, treatment protocols, observer variability, novelty, and statistical justification has been invaluable in strengthening the scientific rigor of our work. We have carefully addressed each concern and implemented substantive revisions throughout the manuscript. Below, we provide a detailed point-by-point response to each comment, as detailed in the attached document.

Reviewer 2 Report

Comments and Suggestions for Authors

This manuscript presents some interesting clinical data comparing the ability of MRI based assessment of tumor response to pathologic response in patients with advanced rectal cancer following neoadjuvant chemoradiotherapy. The authors findings were consistent with previous literature, showing relatively high specificity and low sensitivity in predicting for specific pathologic outcomes. The major concerns are with the novelty and significance of these findings in light of previous similar studies, and the appropriate application and interpretation of statistical methods.

The novelty of this study and its findings is not clearly expressed with respect to previous studies. This should be better explained. 

Low sensitivity, especially for detection of complete response, suggests identification of remaining viable tumor is being assessed more conservatively. High specificity is likely clinically more desirable in order not to under treat.

Cohen's Kappa is designed for measuring agreement between two raters using the same set of categories, so it is stricter than correlation. In this case there are two ordered grading scales whose categories do not strictly align. The Kappa value should be interpreted with caution. Other options for association, like Spearman's, or an approach for ordinal association measure like Kendall’s τ-b or Somer's D may be more appropriate or useful as additional measures.

From Table 6 it appears that mrTRG ratings are more compressed (ie most are rated as 2 or 3) as compared to pTRG. The distribution of mrTRG ratings does appear to move in the right direction as pTRG rating increases. Perhaps a regression model could be used to map mrTRG onto pTRG to improve the predictive ability.

Adding the total count and % to Table 6 would be helpful. Adding a figure to visualize the contingency table might also be helpful to understand the data (heatmap or other plot).

Author Response

We sincerely appreciate the thorough and expert critique from Reviewer 2 regarding the novelty of our findings, our statistical methodology, and data presentation. The insightful comments on positioning our work within the context of previous literature, the appropriate application of ordinal correlation measures, and recommendations for enhanced visualization of our contingency table have significantly strengthened both the scientific rigor and clarity of our manuscript. We have carefully addressed each point with substantive revisions throughout. Below, we provide detailed responses to each comment, as outlined in the attached document.

Reviewer 3 Report

Comments and Suggestions for Authors

Congratulations on the interesting article. In recent years, Total Neoadjuvant Treatment (TNT) has established itself as a mainstay in the treatment of LARC (I'll cite one article that emphasize the role of chemoradiation: PMID: 36765878), achieving excellent results in terms of downstaging, downsizing, and downgrading. The results are sometimes so exceptional that they achieve a complete response and the patient is directed toward NOM (non-operative management). To achieve this, the role of MRI is crucial: it allows us to precisely define the degree of response to treatment, combined with endoscopy.
The implementation of radiomics textures will certainly play a crucial role in the coming years.
Best regards

Author Response

Comment: Congratulations on the interesting article. In recent years, Total Neoadjuvant Treatment (TNT) has established itself as a mainstay in the treatment of LARC (I'll cite one article that emphasize the role of chemoradiation: PMID: 36765878), achieving excellent results in terms of downstaging, downsizing, and downgrading. The results are sometimes so exceptional that they achieve a complete response and the patient is directed toward NOM (non-operative management). To achieve this, the role of MRI is crucial: it allows us to precisely define the degree of response to treatment, combined with endoscopy.

The implementation of radiomics textures will certainly play a crucial role in the coming years.
Best regards

Response:

We sincerely appreciate the reviewer's encouraging comments and thoughtful perspective on our work. We fully agree with the emphasis on the evolving role of Total Neoadjuvant Treatment (TNT) in LARC management and its impact on treatment response assessment and organ preservation strategies.

  • TNT and Non-Operative Management Context:

The reviewer correctly highlights that TNT has become increasingly central in LARC treatment protocols, particularly with the goal of achieving complete clinical response and facilitating non-operative management (NOM) strategies. Our study, while conducted in a period when TNT protocols were less uniformly adopted at our institution, contributes meaningfully to this evolving landscape by providing a comprehensive radiopathological correlation dataset from 97 consecutively treated patients. The findings demonstrate that MRI, although valuable for confirming favorable responses (high specificity), has limitations as a standalone discriminator of complete pathological response. This underscores the importance of multimodal assessment—integrating MRI with clinical examination and endoscopy—precisely as advocated in contemporary TNT-NOM protocols.

  • Radiomics and Future Directions:

We enthusiastically concur that radiomics and textural analysis represent promising frontiers for enhancing MRI-based response prediction. Our systematically annotated imaging dataset from surgically explored patients provides a valuable resource for future radiomics investigations. As organ preservation strategies become more prevalent, opportunities for prospective radiomics validation are diminishing, making our dataset particularly suited for algorithm development and external validation studies aimed at bridging the gap between visual assessment and advanced imaging analytics.

We are grateful for the thoughtful feedback and the opportunity to position our work within the broader context of contemporary LARC management and emerging imaging technologies.

Round 2

Reviewer 1 Report

Comments and Suggestions for Authors

 As mentioned before,  validation of intraobserver error is needed in this kind of study. 

Reviewer 2 Report

Comments and Suggestions for Authors

The authors have satisfactorily addressed my concerns, and I would recommend publication with no addition comments.